# Biofilms in Chronic Wound Infections: Innovative Antimicrobial Approaches Using the In Vitro Lubbock Chronic Wound Biofilm Model

**DOI:** 10.3390/ijms24021004

**Published:** 2023-01-05

**Authors:** Firas Diban, Silvia Di Lodovico, Paola Di Fermo, Simonetta D’Ercole, Sara D’Arcangelo, Mara Di Giulio, Luigina Cellini

**Affiliations:** 1Department of Pharmacy, University “G. d’Annunzio” Chieti-Pescara, 66100 Chieti, Italy; 2Department of Medical, Oral and Biotechnological Sciences, University “G. d’Annunzio” Chieti-Pescara, 66100 Chieti, Italy

**Keywords:** chronic wound, Lubbock chronic wound biofilm model, *Pseudomonas aeruginosa*, *Staphylococcus aureus*

## Abstract

Chronic wounds have harmful effects on both patients and healthcare systems. Wound chronicity is attributed to an impaired healing process due to several host and local factors that affect healing pathways. The resulting ulcers contain a wide variety of microorganisms that are mostly resistant to antimicrobials and possess the ability to form mono/poly-microbial biofilms. The search for new, effective and safe compounds to handle chronic wounds has come a long way throughout the history of medicine, which has included several studies and trials of conventional treatments. Treatments focus on fighting the microbial colonization that develops in the wound by multidrug resistant pathogens. The development of molecular medicine, especially in antibacterial agents, needs an in vitro model similar to the in vivo chronic wound environment to evaluate the efficacy of antimicrobial agents. The Lubbock chronic wound biofilm (LCWB) model is an in vitro model developed to mimic the pathogen colonization and the biofilm formation of a real chronic wound, and it is suitable to screen the antibacterial activity of innovative compounds. In this review, we focused on the characteristics of chronic wound biofilms and the contribution of the LCWB model both to the study of wound poly-microbial biofilms and as a model for novel treatment strategies.

## 1. Introduction

A chronic wound is a wound that does not heal in the right order nor quickly enough in order to achieve the functional integrity of the skin. Impaired healing can lead to microorganism colonization of the wound bed resulting in exudate production and pain [1]. Normal wound healing is a complex process that involves specific stages, summarized in the following (Figure 1):(A)Coagulation and hemostasis: blood vessels contract and the coagulation cascade seals the injured area and minimizes blood loss [2];(B)Inflammation: neutrophils migrate to the damaged area by the chemo-attractants, especially platelet factor 4 (PF4) and interleukin (IL)-8 (IL-8). Neutrophils start the debridement by engulfing and killing bacteria through production of reactive oxygen species (ROS) and proteases [3,4];(C)Proliferation: angiogenesis, granulation tissue formation and keratinocytes re-epithelialization are the main features of this phase [5,6,7,8];(D)Remodeling: in this final phase, the wound continues the healing activity and extracellular matrix (ECM) maturation to achieve tissue integrity and function [2,9,10].

There are several types of chronic wounds, including venous ulcers, arterial ulcers, diabetic foot ulcers and pressure ulcers which, in particular, have been estimated to affect between 1 and 3 million people per year in the United States [11]. Approximately 80% of patients with venous ulcers suffer from pain with a mean intensity of four (on a scale of 0–10) [12]. Taking into consideration the “chronicity” of this case and the need for intensive care (at home or hospital), in addition to the previous mentioned symptoms, quality of life is severely affected [13]. Chronic wound development is attributed to several factors including comorbidities (diabetes and obesity), medications (non-steroidal anti-inflammatory drugs, NSAIDs, and steroids), oncology interventions (chemotherapy and radiation) and some habits (smoking and alcoholism) [14]. As injury occurs, the skin barrier loses its defensive function and the wound area becomes exposed to various pathogens paving the way for microbial colonization of the wound site. The level of microbial colonization and the presence of persistent infection play an important role in wound healing and its chronicity. Antibiotic-resistant infections associated with the presence of biofilm lead to a worldwide economy loss and over a half million deaths annually [15]. The new challenges are focusing on tackling the poly-microbial biofilm in chronic wounds by using sustainable approaches that include non-antibiotic compounds in line with the “One Health” approaches.

Therefore, the aim of this review is to study the role of mono- and poly-microbial biofilms in chronic wounds and to study an innovative in vitro model, the Lubbock chronic wound biofilm, as a suitable 3D gradient to realize the microbial spatial distribution in a human-like chronic wound environment. In addition, this review focusses its attention on new sustainable approaches to counteract the chronic wound pathogens in biofilms.

## 2. Clinical Significance of Chronic Wounds

### 2.1. Microbial Composition in Wound Site

Microbial colonization plays an important role in terms of the significance of the chronic wounds. In fact, microbial multiplication is a critical step in the “chronicity” of the wound because the immune response is unable to fight the infection, with the microorganisms often organized into mono- or poly-microbial biofilms. The wounds microbial colonization involves different aerobic and anaerobic pathogenic microorganisms including bacteria and yeasts [16,17]. Among the pathogens, *Staphylococcus aureus, Pseudomonas aeruginosa, Candida albicans* and β-hemolytic streptococci are the primary causes of delayed wound healing and infection. Wu et al. showed the prevalence of more frequent bacterial strains in patients with diabetic foot ulcers (different stages depending on Wagner grade classification). The authors demonstrated that the ulcers were infected by single strains (56.8%) or multiple pathogens (43.2%) and the type of microorganisms were mostly Gram-negative bacilli and Gram-positive with different distributions. More severe ulcers tend to harbor Gram-negative bacilli to a greater extent [18]. Other studies on microbial distribution have demonstrated the predominance of *S. aureus* and *P. aeruginosa* presence in chronic wounds [19,20,21,22]. Moreover, Dowd et al. studied the prevalence of yeasts in chronic wound samples, with 23% of specimens in the survey testing positive for yeasts, mostly of the genus *Candida* [23]. In general, the bacterial distribution in the chronic wound bed suggests that *P. aeruginosa* tends to settle in the deeper part of chronic wound bed, forming clusters inside the self-produced extracellular polymeric matrix substance (EPS), while *S. aureus* colonizes in the upper layer. This different distribution leads to variations in culturing results depending on the sampling technique (superficial swab or deep sample) [16,24,25,26].

Besides *S. aureus* and *P. aeruginosa*, a wide diversity of bacterial strains have been detected in chronic wound sites by culturing or molecular detection methods. Studies have indicated the involvement of several bacterial species including *Enterococcus faecalis*, anaerobic bacteria (such as *Finegoldia* spp. and *Anaerococcus* spp.), *Proteus* spp. and β-hemolytic Streptococci. Furthermore, commensal bacteria such as coagulase-negative *Staphylococcus* (*S. epidermidis*) and *Corynebacterium* spp. have been isolated from chronic wound samples [22,27,28].

Transcriptomic analyses of different bacterial species showed that the existence of these bacteria in chronic wounds seems to enhance the expression of specific genes responsible for increasing the antibiotic resistance and virulence factors [26]. Antimicrobial resistance is attributed to several mechanisms including enzyme production, efflux pumps and acquired genes that provide resistance against antimicrobials [29]. Bessa et al. have showed that there has been an increase in wound bacteria resistant to different antibiotics. In particular, Gram-negative bacteria showed a high resistance to most antibiotics [30]. Researchers have highlighted another scenario that plays a crucial role in antimicrobial tolerance: the presence of persister cells. Persisters are defined as a phenotypic variant of bacteria that cause a reduction in the metabolic activity and growth rate, providing the ability to survive antimicrobial treatment. *Escherichia coli*, *Salmonella* spp., *P. aeruginosa* and *S. aureus* have the ability to form persisters [31]. To overcome this problem, researchers have screened the efficacy of antibiotics that can kill persisters using growth-independent molecules such as daptomycin and mitomycin C [31,32].

### 2.2. Biofilms in Chronic Wounds

It is known that a great percentage of chronic wound microorganisms tend to develop biofilms that are defined as “structured consortiums of aggregated microbial cells, surrounded by a polymer matrix, that adhere to natural or artificial surfaces” [33]. Biofilm formation is dynamic and consists of several steps: adhesion to a surface; proliferation, resulting in the formation of a microcolony; and growth and differentiation. Developed biofilms have characteristic structures such as mushroom towers and water channels that carry nutrients and water to, and wastes away from, the lower layers of the biofilm. The major part of the EPS matrix in biofilms is composed mainly of substances produced by microorganisms such as exopolysaccharides, extracellular proteins, extracellular DNA and lipids. Extracellular polymeric substances facilitate adhesion and aggregation, stabilize the biofilm cellular component, provide an interactive environment between bacteria and provide a protective barrier against the host immune system [34]. Finally, the microcolonies grow larger until the development of a mature biofilm that releases new planktonic bacteria to the surrounding environment [35]. The matrix of the microbial biofilm protects the microorganisms from the antibiotics, avoiding drug penetration at bactericidal concentration. In addition, the transfer of antibiotic resistant genes among bacteria is realized in the biofilm, increasing the antimicrobial tolerance. The reduced metabolic activity of sessile bacteria within biofilms appears to hinder the antimicrobial activity, since these agents require bacterial growth to be effective [29,36]. Therefore, infections associated with microbial biofilms are associated with a high degree of recalcitrance due to the interaction of antimicrobials with the components of the biofilm matrix, the reduced growth rates and the various actions of specific genetic determinants of antibiotic resistance and tolerance [37]. In addition, the production of many virulence factors, including biofilm formation, have been attributed to cell-to-cell communication via quorum sensing (QS). This is a system responsible for cell-to-cell communication among microorganisms and is controlled by signaling molecules that bind to response regulators and, as a result, modify the transcription of multiple genes involved in encoding bacterial virulence factors. In fact, bacteria produce extracellular signals during the adaptation phase, and once a critical cell density threshold is reached, these signals interact with their related receptors and coordinate the expression of associated genes. Several QS systems have been identified in Gram-positive and Gram-negative bacteria, including acyl-homoserine lactone (AHL) and peptide-based QS systems. AHL QS is necessary for bacteria adaptation, cellular growth, cell adhesion, biofilm development, cell division, antibiotic resistance, plasmid conjugation and virulence gene expression in Gram-negative bacteria [38]. In *P. aeruginosa*, the QS was found to have an important role in biofilm formation by generating extracellular DNA which is regulated by a *Pseudomonas* quinolone signal (PQS)-based quorum-sensing system. Additionally, biosurfactant rhamnolipid, LecA and LecB lectins and siderophores (pyoverdine and pyochelin) are important for biofilm formation and all are regulated by quorum-sensing. All these features indicate the importance of QS as a possible target of antibiofilm agents to interrupt the biofilm formation and increase antimicrobial treatment efficacy [38].

Fungal species express the ability to form biofilms in a similar way but with different structural compositions. In general, fungal biofilms contain protein, carbohydrate, lipid and DNA. *Candida albicans* has the ability to produce biofilms with polysaccharide units similar to its cell wall structural polysaccharides, but with a more interconnected and rich texture that participates in the antifungal tolerance [39].

Most chronic wounds are characterized by poly-microbial biofilm (60%) inside the wound bed in comparison to acute wounds (6%) [40]. In a poly-microbial biofilm, the microorganisms establish interspecies interactions, allowing for the exchange of genetic information, and exhibit synergistic or competitive relationships leading to increased antimicrobial tolerance [41]. In chronic wounds, biofilm formation leads to several symptoms including pale and edema wound bed, fragile granulation tissue, large amount of exudate, necrotic and rotting tissue, wound pain and a pungent smell [42]. These clinical features may serve in the diagnosis of biofilms but lack accuracy for diagnosis. In addition to electron microscopy for biofilm observation, several developed approaches can be used in this context, including wound blotting, staining the EPS from the biofilm, detection of bacteria (or biofilm components) by use of fluorescent probes and detection techniques to monitor the bacterial fluorescence response to violet light [43].

Nowadays, the aims of different works are to better understand the poly-microbial biofilm mechanisms and to discover new therapeutic strategies to act, for example, on the QS system that can represent novel anti-virulence approaches. Quorum sensing inhibitors (QSIs) are derived from various origins and operate by several mechanisms to suppress QS at different levels of signaling pathway. QSIs have the ability to inhibit the synthesis of signal molecules and also to degrade and compete with them, leading to the disruption of biofilm formation mediated by QS [44]. In particular, phenolic compounds or other bioactive compounds are able to affect the QS and therefore reduce the biofilm formation [45,46].

## 3. In Vitro Wound Models

The search for new strategies to counter the development of antibiotic resistance is a major global challenge for the life sciences community and for public health. Wound models are an in vitro useful representation of the realistic chronic wound, including the factors that interfere with the healing process. The diversity of factors affecting this complex process leads to challenges in its simulation. Nonetheless, different in vitro wound models have been developed to test the efficacy of antimicrobials and antiseptics and to test novel approaches to counteract microorganisms and biofilms involved in wound chronicity.

A recent review has detailed the in vitro models used in this field. These models should manifest specific characteristics to be as similar as possible to a real wound environment, including wound simulating media (WSM), host matrix, the continued addition of nutrients, several specific species and the 3D gradients [47]. One example of these models is the human plasma biofilm model (hpBIOM) which was developed by combining human plasma with a buffy coat of the same donor followed by addition of the targeted biofilm forming pathogen. Moreover, clot formation occurs by the effect of calcium chloride to obtain the fibrin polymerization and the final coagulation. This model takes into account the human immune system role which was absent in other in vitro models [48]. Several studies have been conducted using hpBIOM to evaluate the antibiofilm activity of conventional antimicrobial compounds and novel therapies [49,50,51].

Chen et al. developed a novel chronic wound biofilm model (layered chronic wound biofilm model) based on built-in layers of nutrients (peptone, serum and blood) combined with 0.5% agar to form a double layer (fat and dermis layers) in the bottom of each well of a 4-well plate. These layers mimicked the dermis tissue in human skin. Then, two layers of the selected bacterial strains were placed on top, taking into consideration the previously described bacterial distribution in a real chronic wound, in which *S. aureus* is located on the surface while *P. aeruginosa* is in the deeper part. This model provided a suitable environment to grow a dual species microbial biofilm that lasts for 96 h, allowing the application of tested antimicrobial agents (solutions and wound dressings) against 48 h biofilm. Lastly, the treated biofilms were easily harvested and prepared for counting microbial reduction, live/dead tests and SEM observation [52].

## 4. Lubbock Chronic Wound Biofilm (LCWB)

### 4.1. LCWB Models

The Lubbock chronic wound biofilm (LCWB) model is a versatile in vitro wound model that easily reproduces a chronic human-like wound. It is an inter-kingdom biofilm that mimics the realistic microbial spatial proliferation in wounds. This in vitro model is widely recognized to more closely resemble the in vivo human wound environment for the wound simulating medium, host matrix, several chosen species, 3D gradients, flow and growth on a solid surface [47]. The LCWB contains the main pathogens isolated from a chronic wound that are able to form a multispecies biofilm similar to the realistic wound biofilm. Moreover, the combination of Bolton broth with 50% plasma and 5% freeze–thawed horse red blood cells gave a nutritive mixture similar to the chronic wound components. Damaged tissue, red blood cells and serum resembled the wound environment that promotes bacterial growth and biofilm formation. Sun et al. developed this model by incubating *P. aeruginosa*, *S. aureus* and *E. faecalis* with the nutritive mixture for 24 h using glass test tubes [53]. The multispecies bacterial environment in LCWB put all strains involved under mutual effects which could lead to imbalance in the bacterial composition of the media. It is known that *P. aeruginosa* products influence *S. aureus* growth in planktonic co-cultures due to several virulence factors, pyocyanin and 4-hydroxy-2-heptylquinoline N-oxide (HQNO), which both inhibit oxidative respiration of *S. aureus*, iron chelating siderophores such as pyoverdine and pyochelin and lasA protease (staphylolytic specific endopeptidase), causing *S. aureus* cell lysis and cis-2-decenoic acid and rhamnolipids which stimulate biofilm dispersal [54].

In the LCWB model, the environmental conditions in situ influence the poly-microbial growth. Smith et al. suggested that albumin existing in the plasma contributes to the protection of *S. aureus* from *P. aeruginosa* compared to the co-culture of both strains in the planktonic form grown in other medium [55]. Moreover, the co-existence of *S. aureus* and *P. aeruginosa* may alter the response to antibiotics. DeLeon et al. examined this point and concluded that the combined bacterial culture of *S. aureus* and *P. aeruginosa* in the LCWB model remarkably enhanced the antibiotic tolerance compared to the single planktonic culture of each strain. This increased tolerance is attributed to both the host-derived matrix (coagulated medium) and EPS produced by bacteria [56]. Additionally, Dalton et al. tested the properties of an in vivo infected mouse model using bacterial strains grown in vitro in the LCWB model. The wounded mice developed chronic wounds in which all bacterial strains were involved in a poly-microbial biofilm, resulting in impaired wound healing and increased antimicrobial tolerance (in comparison to the mono species biofilm of each bacteria) [57]. Similar results were concluded in a study by Klein et al., in which the researchers infected a LCWB porcine wound model. The poly-microbial biofilm wounds exhibited sustained inflammation and a reduction in healing rate compared to the controls [58].

The LCWB model was used in screening studies regarding biofilm forming pathogens. Some researchers used a nutritive mix, also named wound-like medium (WLM), to grow a mono species microbial biofilm [59], while others managed to study the poly-microbial biofilm species by inoculating different bacterial strains into the model, including *S. aureus*, *P. aeruginosa*, *E. faecalis* and *B. subtilis*. Moreover, this last model was subjected to some modification in order to test solid antimicrobial dressings. This modified LCWB model consisted of Bolton broth, 1% gelatin, 50% porcine plasma and 5% freeze–thawed porcine erythrocytes in polystyrene tubes, and were inoculated with the standardized bacterial species and incubated for 48 h. Then, the biofilm was moved to an artificial wound bed composed of Bolton broth supplemented with 1% (*w*/*v*) gelatin and 1.2% (*w*/*v*) agar (Figure 2). In every case, the final result was a complex EPS embedding the bacterial cells of each species [60].

As the LCWB model was used to evaluate the effect of different therapeutic treatments on bacterial infections similar to chronic wound infections, other studies exploited the WLM as a basic structure to test bacterial characteristic changes. Pouget et al. modified the composition of a LCWB model and added 10% glucose to better mimic the environment of DFU chronic wounds. In detail, the researchers used WLM in the evaluation of the phenotype and virulence switch of *S. aureus* when incubated with 10% glucose and two antibiotics (vancomycin and linezolid). This stressing condition increased the development of small colony variants of *S. aureus* and decreased the expression of virulence factors [61]. Pouget et al., in a different study, provided another modification to the LCWB model. The change in composition was applied to make the medium (named chronic wound medium, CWM) more similar to a human chronic wound by using human blood derivatives, adding a cellular component and adjusting the pH. In detail, 0.5% of hemolyzed human blood and 20% of heat-inactivated human serum with 79.5% of Bolton broth were used in this model. Additionally, the researchers added human keratinocyte debris to mimic the cellular and inflammatory situation, and controlled the pH of the model by using a HEPES/NaOH buffer to reach a pH of 8 [62]. In conclusion, the CWM model provided a suitable environment that facilitated bacterial adhesion and biofilm formation and can be considered as a reproducible and reliable medium to study chronic wound infections [62]. In another study, Pouget et al. combined the CWM model with BioFluxTM microfluidic system to study and observe the biofilm formation with a fluorescence inverted microscope [63].

### 4.2. LCWB as Model for Detection of Novel Strategies

The LCWB model, a well-structured multispecies biofilm, was used to evaluate the activity of non-antibiotic compounds and novel techniques as possible antibacterial, antibiofilm approaches to chronic wound healing treatments (Table 1, Figure 3). As shown by Di Lodovico et al., the *Capparis spinose* aqueous extract significantly reduced the biofilm biomass and the colony forming unit/mg of bacteria and inhibited the virulence factors of *P. aeruginosa*. *Capparis spinose* aqueous extract reduced the LCWB by 97.32% ± 2.29 and 99.67% ± 0.07 for resistant *S. aureus* for *P. aeruginosa* strains, respectively [64]. In another study, Di Fermo et al. proposed an inter-kingdom poly-microbial LCWB, demonstrating the modulating action between bacteria and yeasts of a new peptide, L18R. This peptide showed antimicrobial activity against all strains in planktonic form (especially *C. albicans*), affecting biofilm formation of *C. albicans* with less effect against the bacterial strains [65]. The modulating action of L18R against bacteria and yeast is an important topic in chronic wound management in association with conventional antibiotics or alternative treatments.

The Lubbock chronic wound biofilm was also used to test the antimicrobial proprieties of graphene oxide (GO), since GO is known for its antibacterial properties and it has been tested against multidrug-resistant bacterial strains [73]. Two studies engaged GO in the experiments of antimicrobial efficacy in a LCWB model. The first applied an aqueous solution of GO (50 mg/L) to a LCWB model in the early and late stages of biofilm formation. The obtained results indicated the important role of GO in reducing microbial growth in both situations of early and late stages; disrupting the fibrin network in the biofilm structure. The authors demonstrated the wrapping effect of GO against the bacteria, reducing the CFU/mg by up to 70.24% ± 4.47 and 59.31% ± 16.84 for *S. aureus* and *P. aeruginosa,* respectively [68]. The second study highlighted the combined action of GO and photodynamic therapy PDT as a non-antibiotic methodology in wound biofilm treatment. Results showed the antimicrobial effect of this combination against resistant bacteria and poly-microbial biofilms by the multiple effective factors derived from GO antibacterial activity and the ROS produced by PDT and irradiated GO [71]. The use of eco-friendly light emitting diodes (LEDs) increased the action of GO against *P. aeruginosa,* reaching a reduction of 95.17% ± 2.56 with respect to the control. Another study exploited WLM in the preparation of in vitro biofilms of *P. aeruginosa* in order to compare the mechanical properties of biofilms both in vitro and in vivo. The results indicated the differences between in vitro and in vivo biofilm mechanical properties, probably caused by the distinct environmental conditions in both situations in terms of surface attachment, growth rate, temperature, pH or nutrient availability [74].

Pirlar et al. studied the joint effect of enzymatic targeted treatment against the dual species biofilm of *S. aureus* and *P. aeruginosa* in LCWB models. The experimental study utilized three different enzymes (trypsin, β-glucosidase and DNase I) specifically targets at the main components of EPS of the biofilm (proteins, polysaccharides and extracellular DNA). The enzymes were applied alone and in combination with each other or with antibiotics. As a result, the enzymatic treatment (trypsin, β-glucosidase and DNase I) was able to disrupt the bacterial biofilm in all conditions with minimum effective concentrations of 1 μg/mL, 8 U/mL and 150 U/mL, respectively, and the best combination was between trypsin and Dnase I (0.15 μg/mL and 50 U/mL). Moreover, the enzymes reduced the minimum biofilm eradication concentration (MBEC) of meropenem and amikacin needed to affect the biofilm. The study suggested the possible use of the enzymatic mixtures for disinfecting surfaces and medical devices [75].

## 5. Understanding Biofilms Using the “Zone Model”

In general, the LCWB model resembles a mono- or poly-microbial biofilm and mimics the colonization of a chronic wound. However, it is missing a crucial factor in the immune system counterpart, the immune system representation. The importance of immune cells starts from the very first step in wound formation through each step of wound healing, as previously described. Furthermore, it is hard to detect the interaction between white blood cells, inflammatory cytokines and the biofilm structure, limiting its overall representation of real chronic wound characteristics. Therefore, the LCWB model, as an in vitro wound model, is a suitable model for primary screening of novel treatments against microbial biofilms (whereas studying the immune system role needs in vivo animal models to detect) [76].

A recent review has introduced a new perspective on the bacterium–host interactions and the levels of this relationship. The “zone model” illustrates five distinguished zones of the biofilms of chronic wounds; each of them interacts with the other and displays distinctive characteristics that simultaneously represent the wound environment. More accurately, the zones are arranged starting from the single bacterium, the bacterial aggregates in the matrix, the surrounding environment, the wound tissue and, finally, the host itself. The researchers suggested that any in vitro model used to study the microbial biofilms in chronic wounds should involve the five zones to better mimic real interactions eventually leading to more realistic and reliable outcomes [77].

From a practical point of view, the in vitro wound model should contain blood components, allowing the formation of mature (mono- or poly-microbial) biofilms and providing a simple method to harvest and quantify the grown biofilms to assess their phenotypical, genetic or any other characteristics [78].

## 6. Promising Therapies in Chronic Wounds

All studies in the field of chronic wound biofilms concern mainly the final purpose: treatment. It is known that chronic wound biofilms are difficult to diagnose in the first place, in addition to characterization of the poly-microbial colonization diversity, antimicrobial tolerance and other hurdles that prevent the establishment of a unified, evidence-based guideline to choose a treatment procedure. Nevertheless, Schultz et al. agreed a consensus document, in which they put important definitions to better describe chronic wound biofilms. Additionally, they indicated debridement, antiseptic agents and antibiotic applications in the treatment plan of chronic non-healing wounds [78].

Numerous therapeutic approaches have been used in wound treatments, including conventional and novel treatments. Novel nanotechnology-based approaches aim to specifically target the structure of the biofilm by facilitating the biofilm penetration and behave similar to a delivery system carrying antimicrobial molecules. Several types of nanoparticles (polymeric, liposomal, metal-based or carbon-based) have been tested regarding their antibiofilm efficacy [15].

-Ultrasonic debridement

A crucial step is the debridement of non-vital tissues followed by the application of a suitable wound dressing to cover the injured area. Furthermore, other methods are used to facilitate the wound healing process such as skin substitutes, negative pressure wound therapy, growth factors and hyperbaric oxygen [79].

Mori et al. have showed the effect of ultrasonic debridement as a biofilm-based wound care treatment in two separate studies. Ultrasonic debridement was applied on different types of chronic wounds of patients both in hospital and home care. These results showed a significant effect of the debridement in the wound healing process [80]. Studies reported that performing debridement in scheduled intervals effectively assists in breaking the resistant structure of the biofilm matrix revealing the embedded microorganisms. This debridement offered a “therapeutic window” that improved the selective targeting of the bacteria with antibiotics [25,81].

-Antiseptics and Antibiotics

Antiseptic application is one important step in wound treatment due to their role in reducing the pathogenic microbial load in the wound site. Povidone-iodine, polyhexanide and silver products have been studied extensively in wound care applications, showing that they perform an effective killing action against a wide diversity of microorganisms (including bacteria and fungi) and the biofilms formed in wounds, with some variations in cytotoxicity and development of resistance [82].

Systematic or local application of antibiotics has been the main part of the treatment plan for chronic wounds. All classes of antibiotics have been used including beta-lactams, aminoglycosides, macrolides, quinolones, lincosamides, nitroimidazoles and selective sulfonamide agents, along with other topical antibiotics [83]. Topical antibiotics may have some advantages over systematic ones, such as the high concentration in the wound bed, limited amount of the antibiotic and limited systematic absorption and toxicity, the possibility to apply novel compounds directly to the wound and the easy application for non-hospitalized patients [84]. The antibacterial efficacy is related to the concentration in the wound bed, which, in this case, could be reduced due to the ischemia and decreased blood flow to the wound site and the protective barrier of EPS in bacterial biofilms. Sub-inhibitory concentrations of antibiotics might lead to development of antimicrobial resistance, narrowing the future medical antibiotic choices [83]. The emerging resistance against antibiotics indicates the importance of a specific, targeted selection of antibiotics in chronic wound treatment [85].

-Antimicrobial peptides

Antimicrobial peptides (AMPs), defined as “oligopeptides with a varying number of amino acids” from natural or synthetic origin, have a broad spectrum of targeted organisms ranging from viruses to parasites. AMPs have different types depending on the secondary structure (α-helix, β-sheet, loop and extended peptides) and have a rapid killing effect by targeting the lipopolysaccharide layer of a cell membrane. Moreover, the combination of AMPs and antibiotics increases the activity of the antibiotics by synergistic effects [86].

AMPs normally exist in the human skin and play a crucial role in enhancing the immune response to bacterial infection. There are several types of identified AMPs, including cathelicidin LL-37 and defensins, which are produced by different immune cells (keratinocytes, macrophages, neutrophils, mast cells and dendritic cells) [87].

In the case of chronic wounds, AMP utilization against poly-microbial infection represents a promising aspect, not only for the broad spectrum of antimicrobial action, but also for the possible destruction of persister cells enclosed inside the biofilm matrix. Furthermore, AMPs exert multiple effects that contribute to the healing process, such as the angiogenic role, wound healing and immunomodulatory activities.

On the other hand, AMPs may have possible weaknesses that could hinder or lower the expected outcome, including the susceptibility to protease degradation, their sequestration by biological fluids, their inactivation by physiological concentrations of salts, and their potential toxicity towards eukaryotic cells [88]. AMPs were studied extensively to evaluate their potential action in fighting antimicrobial resistance in several pathologies. For instance, Maisetta et al. studied a semi-synthetic peptide (Lin-SB056-1) against *P. aeruginosa* in a cystic fibrosis-like medium. The peptide showed a strong bactericidal activity against the planktonic form of *P. aeruginosa*, reduced the biomass of mature biofilm and inhibited biofilm formation (in combination with EDTA) [89].

-Photodynamic therapy

Photodynamic therapy, PDT, is considered a novel and sustainable approach to treat several pathologies including cancer, bacterial colonization and biofilms in dental infections and chronic wounds. It depends on the application of a photosensitizer, PS, compound topically on the target followed by irradiation at an appropriate wavelength. PSs accumulate in the mitochondria, which are responsible for energy production and, after being irradiated, lead to production of ROS, causing cell death [90].

Nesi-Reis et al. reviewed the role of PDT in the treatment of wounds. The authors showed six studies on chronic wounds and the results showed PDT as a promising approach not only for decreasing the bacterial load, but also for its immunomodulatory function and its contribution to re-epithelialization of the wound [91].

-Biodegradable bacterial by-products

Polyhydroxyalkanoates, PHAs, are biodegradable eco-friendly polymers produced by certain bacteria under stress conditions. In this situation, bacteria produce PHAs from carbon sources (bio-wastes) as an energy reserve (mostly as poly-β-hydroxybutyrates PHBs). PHAs have showed promising antimicrobial and antibiofilm activity against several microbial species including those involved in chronic wound biofilm formation. Besides their antimicrobial properties, PHAs exhibit biocompatibility and efficacy in drug delivery and tissue engineering, suggesting their important role in chronic wound management [92].

## 7. Conclusions

In conclusion, the LCWB model offered a convenient and reliable in vitro model to study microorganisms in an environment similar to a chronic wound, with similar host matrix components and spatial microbial distribution. Despite the absence of immune system components in this model, it has been successfully employed in the screening tests of antimicrobial agents and methods. The flexibility of the LCWB model has allowed for several modifications to develop its environment to better represent the physiological conditions found at an infection site. In wound models, it is difficult to achieve an ideal model that covers all features of real chronic wounds. Yet, the LCWB model has passed through several changes from the first model [53], prepared to test liquid materials, to the next model [60] for testing wound dressings loaded with active compounds, to the last established model [62] which displayed more similar characteristics to a human wound state.

## Figures and Tables

**Figure 1 ijms-24-01004-f001:**
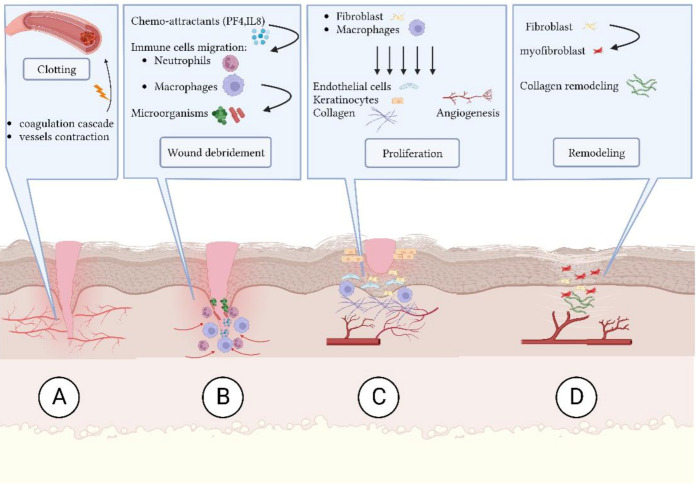
Steps in wound healing process: (**A**) clotting and hemostasis; (**B**) inflammation and debridement; (**C**) proliferation; and (**D**) remodeling. Created with BioRender.com (accessed on 21 October 2022).

**Figure 2 ijms-24-01004-f002:**
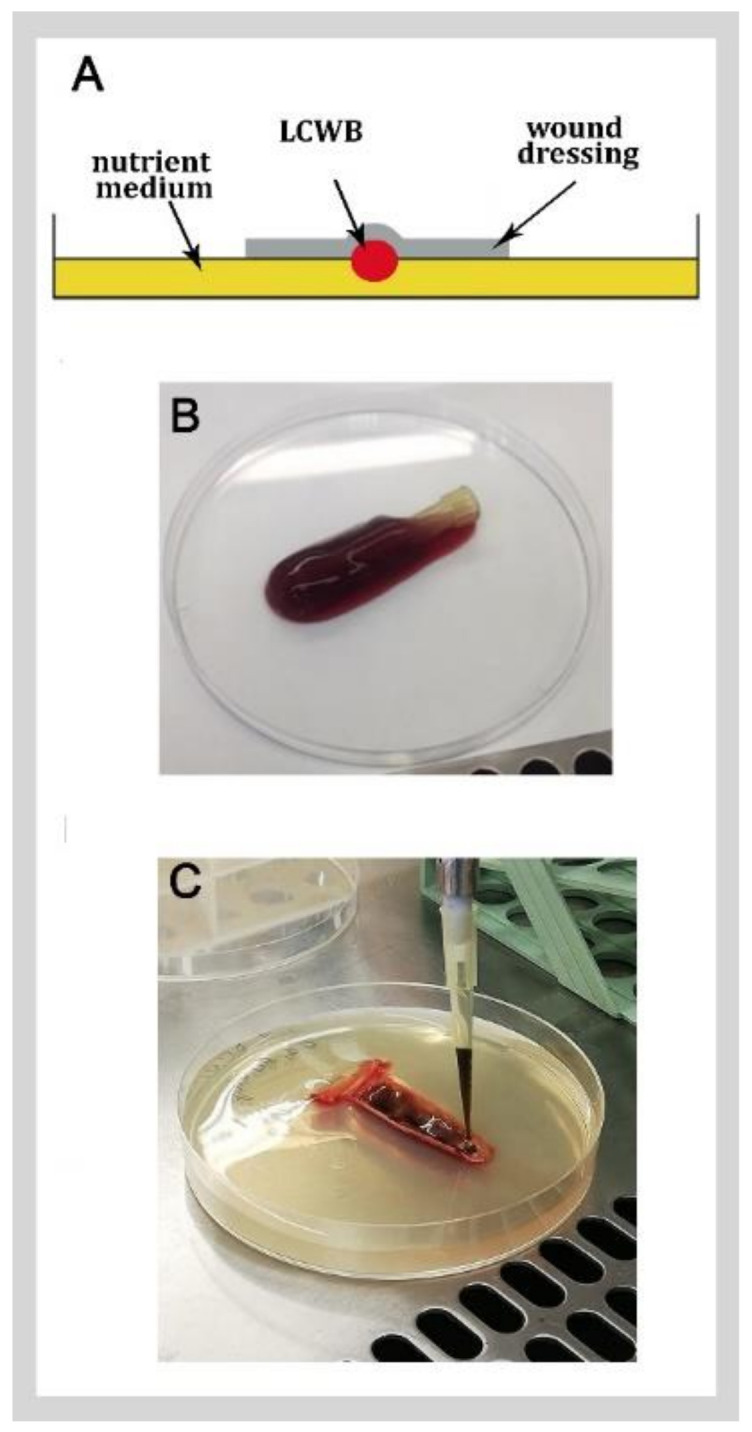
Lubbock chronic wound biofilm model: (**A**) schematic representation of the treatment of LCWB transferred to artificial wound bed, adapted from Kucera et al. [60]; (**B**) mature LCWB of *S. aureus* and *P. aeruginosa* grown together; (**C**) antimicrobial application in a LCWB model located in an artificial wound bed.

**Figure 3 ijms-24-01004-f003:**
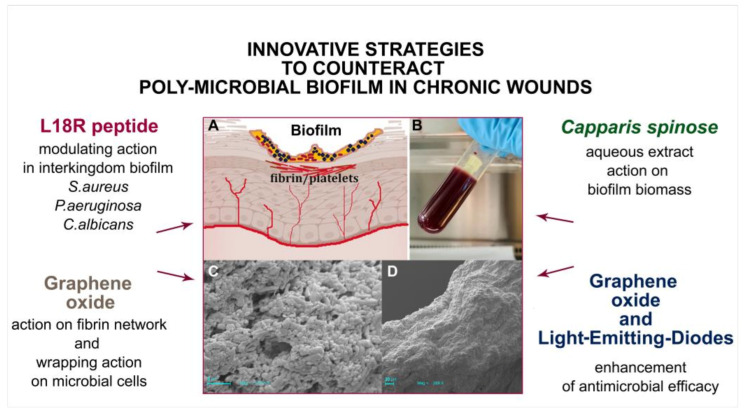
Recent innovative strategies to counteract biofilms in chronic wounds. (**A**) Representative schematization of biofilm in a wound bed; (**B**) the 48h mature biofilm in Lubbock chronic wound biofilm (LCWB) model; (**C**,**D**) Electron Scanning microscopy representative images of LCWB; (C, scale bar 3 µm.; D, scale bar 20 µm).

**Table 1 ijms-24-01004-t001:** Studies conducted using the LCWB model.

Treatment in LCWB	Pathogens in LCWB	Results	Reference
Bleach (6% sodium hypochlorite) triclosan gallium nitrate	*P. aeruginosa* *E. faecalis* *S. aureus* 24 h incubation	In planktonic cells, 1% bleach solution was required for 100% disinfection, while multispecies biofilm withstood 50% concentration; Triclosan inhibited biofilm formation at 1, 10 and 100 ppm concentrations with a selective inhibitory effect on *S. aureus;* Gallium inhibited biofilm formation at 1 μM with a selective inhibitory effect on *P. aeruginosa.*	Sun et al., 2008 [53]
Anaerobic bacteria integration in LCWB model	*P. aeruginosa* *E. faecalis* *S. aureus*	*Clostridium perfringens*, *Peptoniphilus ivorii* and *Anaerococcus lactolyticus* showed a slight growth in biofilm after 24 h and a greater integration of in biofilm after 48 h; *Peptostreptococcus anaerobius* showed integration in biofilm after both 24 and 48 h; The same result was detected with *Finegoldia magna,* which became dominant biofilm after 48 h.	Sun et al., 2009 [66]
Several biofilm effectors	*P. aeruginosa* *E. faecalis* *S. aureus*	20% xylitol, 10% erythritol, 1000 µg/mL farnesol, 20 mM salicylic acid or 0.1% of either of the two gel formulations were able to inhibit biofilm formation	Dowd et al., 2009 [67]
Polyvinylpyrroli- done–iodine complex Cadexomer–iodine complex	*P. aeruginosa* *E. faecalis* *S. aureus* *Bacillus subtilis* 48 h incubation	Cadexomer–iodine (1.8 mg I_2_/cm^2^) reduced bacterial count in the biofilm (5 log reduction in *S. aureus* and *B. subtilis*, 9 log reduction in *P. aeruginosa* and *E. faecalis*); Both (PVP iodine and cadexomer-iodine) were ineffective at the concentration of 0.2 mg I_2_/cm^2^.	Kucera et al., 2014 [60]
Gentamicin Ciprofloxacin Tetracycline	*P. aeruginosa* *S. aureus*	The poly-microbial growth of these strains showed a synergistic effect regarding the antimicrobial tolerance in LCWB compared to the planktonic culture, showing the role of host derived matrix in anti-microbial tolerance enhancement; The deletion of the ica gene in S.aureus and the algD gene in *P. aeruginosa* decreased the tolerance of the coculture in comparison to the wild-type of both microorganisms.	DeLeon et al., 2014 [56]
*Capparis spinose* aqueous extract	*P. aeruginosa* *S. aureus* 48 h incubation	*Capparis spinose* aqueous extract reduced the LCWB formation by 97.32% and 99.67% for resistant *S. aureus* and *P. aeruginosa* strains, respectively.	Di Lodovico et al., 2022 [64]
Octenidine Dihydrochloride 0.1% Povidone–iodine 10% Chlorhexidine digluconate 0.02%	*S. aureus* monomicrobial biofilm in LCWB medium loaded into a prosthetic vascular graft infection (PVGI) model	All antiseptics demonstrated significant antimicrobial efficacy, decreasing colony counts, with the superiority of Octenidine against *S. aureus* biofilms grown on vascular graft (7 orders of magnitude CFU reduction); Chlorhexidine worked best against *S. aureus* biofilms integrity on glass coverslips and decreased the surface area covered with *S. aureus* from 73.75 to 10.55%.	Staneviciute et al., 2019 [59]
Graphene oxide 50 mg/L	*P. aeruginosa* *S. aureus*	Graphene oxide showed an antibiofilm effect disrupting the fibrin network and reducing the CFU/mg by up to 70.24% and 59.31% for *S. aureus* and *P. aeruginosa,* respectively.	Di Giulio et al., 2020 [68]
Manuka honey 100% Honeydew honey 100% Honey recombinant Defensin-1 (Def-1) 0.1 and 1 mg/mL	*S. aureus*, *Streptococcus agalactiae*, *P. aeruginosa*, *E. faecalis*	Both types of honey reduced cell viability of *S. aureus* (by a 4 log reduction), *S. agalactiae* (5 log reduction) and *P. aeruginosa* (5 log reduction) but showed no effect against *E. faecalis*; Def-1 reduced the viability of *S. aureus* (5 log reduction at both concentrations) and *P. aeruginosa* (2 log reduction at 0.1 mg/mL and 4 log reduction at 1 mg/mL); Def-1 inhibited biofilm formation of *E. faecalis* and *S. agalactiae* at both concentrations.	Sojka et al., 2016 [69]
Hyperbaric oxygen therapy (HBOT)	*P. aeruginosa*, *E. faecalis*, *S. aureus*	Treatment with HBOT showed a slight but significant reduction of the viability of the three bacterial species after 30 and 90 min of application in vitro.	Sanford et al., 2018 [70]
Antimicrobial Peptide L18R 100 µg/ml	*S. aureus* *P. aeruginosa* *C. albicans*	L18R showed antimicrobial activity against all strains in planktonic form (especially *C. albicans);* L18R reduced biofilm formation of *C. albicans* (97.19% and 98.81% reduction in biofilm biomass for early stage and mature biofilms, respectively), with less effect against the bacterial strains; L18R did not affect the dual and triadic poly-microbial biofilms.	Di Fermo et al., 2021 [65]
Graphene oxide (50 mg/L) and 5-aminolevulinic acid (ALAD) mediated photodynamic therapy (PDT) therapy	*S. aureus* *P. aeruginosa*	The most effective combination was graphene oxide (50 mg/L) application followed by ALAD-PDT (630 nm) with reductions in *S. aureus* (78.96%) and *P. aeruginosa* (85.67%); Additionally, the application of graphene oxide (50 mg/L) for 24 h incubation, followed by ALAD-PDT exhibited a CFU/mg reduction in *P. aeruginosa* (95.17%); Other conditions reduced the CFU count with different percentages.	Di Lodovico et al., 2022 [71]
Glycoside hydrolases, GH, (α-amilase and cellulase, to target the EPS of biofilm)	*S. aureus* SA31 *P. aeruginosa* PAO1	After 48–96 h of growth, the dual species biofilm was treated for 1 h with either 1× PBS, 2.5% amylase, 2.5% cellulase, or both (5% GH); Amylase significantly dispersed both *P. aeruginosa* and *S. aureus*; The ability of cellulase to disperse *S. aureus* was completely abated as well as to disperse *P. aeruginosa;* One possible explanation for this result is that the activity of cellulase is inhibited by proteolytic blood components in the microcosm model and ex vivo tissue.	Redman et al., 2020 [72]

## Data Availability

Not applicable.

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
