# Peer review of "Biofilms in Chronic Wound Infections: Innovative Antimicrobial Approaches Using the In Vitro Lubbock Chronic Wound Biofilm Model"

_ijms, 2023, doi:10.3390/ijms24021004_

Round 1
Reviewer 1 Report
The aim of the study was to present Lubbock Chronic Wound Biofilm model as a suitable in vitro model to screen the efficacy of innovative approaches against pathogens causing chronic wound infections. The authors described extensively the principles of the physiological wound healing process, the origin of chronic wounds, their clinical significance, aspects of biofilm in chronic wounds, and the established in vitro wound models with an emphasis on the Lubbock Chronic Wound Biofilm. However, the Authors focused on the very obvious aspect of the chronic wound and therefore quite briefly described the issue listed in the title. Therefore, the text does not sufficiently correspond to the title. Moreover, some of the errors in the manuscript indicate insufficient clinical and/or microbiological proficiency of the Authors in the presented topic.
My concerns are as follows:
· keywords should be listed in alphabetical order, in my opinion,
· Line 106 – rather “Diabetes Mellitus”,
· Line 107 – rather “leukocyte”,
· Lines 130 and 132 – “spp” should end with a dot,
· Line 192, etc. - lack of italics,
· Table 1 – rather “Polyvinylpyrrolidone”
· Figure 2 is of rather poor quality.
· line 401 – “all bacterial strains isolated from wound sites were resistant against cefixime and cefpodoxime application” it is totally unclear for the reader since the authors do not even mention what were the exact bacteria isolated from the mentioned chronic wounds. Meanwhile, some of the previously described pathogens (e.g. Pseudomonas aeruginosa strains) are intrinsically resistant to these cephalosporins. Moreover, Staphylococcus aureus strains are normally also resistant at least to one of them. Last of all, the mentioned antibiotics are available for the oral application only and thus altogether their usefulness for the treatment of chronic wound infections is of limited significance, making the whole paragraph confusing for a particular reader.
· Line 402 – “resistant to”, not “resistant against”
· Line 424 – rather “eukaryotic”
Author Response
RESPONSE TO REVIEWER 1 COMMENTS
- Point 1: The aim of the study was to present Lubbock Chronic Wound Biofilm model as a suitable in vitromodel to screen the efficacy of innovative approaches against pathogens causing chronic wound infections. The authors described extensively the principles of the physiological wound healing process, the origin of chronic wounds, their clinical significance, aspects of biofilm in chronic wounds, and the established in vitro wound models with an emphasis on the Lubbock Chronic Wound Biofilm. However, the Authors focused on the very obvious aspect of the chronic wound and therefore quite briefly described the issue listed in the title. Therefore, the text does not sufficiently correspond to the title. Moreover, some of the errors in the manuscript indicate insufficient clinical and/or microbiological proficiency of the Authors in the presented topic.
Response 1: According to the Reviewer comments, we modified the title and rearranged extensively the MS to make the text consistent to the title and to be more pertinent to the proposed topic of the issue (pag. 1, lines 2-5).
- Point 2:keywords should be listed in alphabetical order, in my opinion,
Response 2: In the new version of the MS, we modified the list of the keywords. (pag.1, lines 34-35).
- Point 3: Line 106 – rather “Diabetes Mellitus”,
Response 3: Thanks for your comment, but according with other Reviewers’ comments, we deleted the sentences containing the word " Diabetes Mellitus "
- Point 4: Line 107 – rather “leukocyte”,
Response 4: Thanks for your comment; according with other Reviewers’ comments, we deleted the sentences containing the word " leukocyte"
- Point 5: Lines 130 and 132 – “spp” should end with a dot,
Response 5: Following the Reviewer comment, we modified accordingly (pag. 4-5, lines 165, 167, 179)
- Point 6: Line 192, etc. - lack of italics,
Response 6: Following the Reviewer suggestion, in all of MS we modified accordingly.
- Point 7: Table 1 – rather “Polyvinylpyrrolidone”
Response 7: Following the Reviewer suggestion, we modified accordingly.
- Point 8: Figure 2 is of rather poor quality.
Response 8: In the new version of the MS we included a new figure 2 with higher resolution.
- Point 9line 401 – “all bacterial strains isolated from wound sites were resistant against cefixime and cefpodoxime application” it is totally unclear for the reader since the authors do not even mention what were the exact bacteria isolated from the mentioned chronic wounds. Meanwhile, some of the previously described pathogens (e.g. Pseudomonas aeruginosa strains) are intrinsically resistant to these cephalosporins. Moreover, Staphylococcus aureus strains are normally also resistant at least to one of them. Last of all, the mentioned antibiotics are available for the oral application only and thus altogether their usefulness for the treatment of chronic wound infections is of limited significance, making the whole paragraph confusing for a particular reader.
Response 9: According to the Reviewer comment, in the new version of the MS we deleted the sentence (pag. 16, lines 541-543).
- Point 10Line 402 – “resistant to”, not “resistant against”
Response 10: Thanks for your comment; according with your previous comment, we deleted the sentences containing the word " resistant against"
- Point 11 Line 424 – rather “eukaryotic”
Response 11: Following the Reviewer comment, we modified accordingly (pag. 16, line 565).
Reviewer 2 Report
This review is very thorough and thoughtful, however there were significant english language mistakes that made it very hard to read and in some places understand. Recommend that an english speaking scientist read and correct english.
Sometimes the way it is written, possibly in translation, means that the meaning of the original paper cited is distorted.
eg. line 162 Page 4, 'including wound blotting by certain dyes.' The blotting itself is not carried out by dyes. The wound is blotted and stained.
line 165-6 Page 4, 'surface attached community exhibits impedance to the antimicrobial therapy due to multiple factors such as the poor penetration of the EPS'
The way it reads suggests the EPS has the ability to penetrate the community or maybe the community penetrates the EPS?
Also Figure 2A has been published elsewhere and would need to get permission to publish it here.
Author Response
RESPONSE TO REVIEWER 2 COMMENTS
- Point 1: This review is very thorough and thoughtful, however there were significant english language mistakes that made it very hard to read and in some places understand. Recommend that an english speaking scientist read and correct english.
Response 1: Thank you very much for your comment. The new version of the MS has been completely revised by a native English person.
- Point 2: Sometimes the way it is written, possibly in translation, means that the meaning of the original paper cited is distorted.
Response 2: Thank you for your comment. We revised the entire MS.
- Point 3: eg. line 162 Page 4, 'including wound blotting by certain dyes.' The blotting itself is not carried out by dyes. The wound is blotted and stained.
Response 3: According to the Reviewer comment, we clarified the sentence (pag. 7, line 274)
- Point 4: line 165-6 Page 4, 'surface attached community exhibits impedance to the antimicrobial therapy due to multiple factors such as the poor penetration of the EPS'. The way it reads suggests the EPS has the ability to penetrate the community or maybe the community penetrates the EPS?
Response 4: According to the Reviewer comment, we clarified the sentence (pag. 5, lines 202-213).
- Point 5: Also Figure 2A has been published elsewhere and would need to get permission to publish it here.
Response 5: Thanks for your comment; we inserted a new Figure 2.
Reviewer 3 Report
The manuscript “Lubbock Chronic Wound Biofilm model: a suitable in vitro model to screen the efficacy of innovative approaches against chronic wound pathogens” is exciting and requires significant improvement before its final publication.
Comments
1. The abstract is less informative. Please elaborate and revise it more precisely.
2. Please add a few illustrations, such role of biofilm in chronic infections, the mechanism of diseases, and preventive strategies.
3. Line 28, section 1 should be the introduction. What are nos 1-4? Please revise and update this section with additional pieces of information in brief also, such as i) biofilm, mechanism, its regulations (quorum sensing) in microbial pathogens, and antibiotics resistance issue for treatment i.e., FEMS Microbiology Reviews 41 (2017) 276-301; Molecules 27 (2022) 7584; ii) strategies for regulation of pathogens by inhibition such as the use of quorum sensing inhibitors and bioactive molecules such as antimicrobial agents i.e., polyhydroxyalkanoate derived materials and related compounds (Scientific Reports (2019) 9, 18160; Biotechnology Advances 37 (2019) 62-90; Bioresource Technology 326 (2021) 124737; and iii) few examples of chronic wound pathogens such as P. aeruginosa, S. aureus, E. faecalis and C. albicans their treatment strategies and worldwide economy loss due to such infections (Frontiers in Microbiology 13 (2022) 832919; Angewandte Chemie International Edition 61 (2022) e202112218).
4. The explanation in the citations should be more quantitatively elaborated in the text i.e, Table 1-based citations.
5. Table 1 can be extended with recent citations and brief quantitative data on inhibition.
6. Please provide additional illustrations on the promising therapies in chronic wounds and their significant key points and disadvantages.
7. Please add a new section about major challenges in chronic wound pathogens, especially antibiotic resistance, and potential strategies to counter them i.e., uses of quorum sensing inhibitors.
Author Response
RESPONSE TO REVIEWER 3 COMMENTS
The manuscript “Lubbock Chronic Wound Biofilm model: a suitable in vitro model to screen the efficacy of innovative approaches against chronic wound pathogens” is exciting and requires significant improvement before its final publication.
- Point 1: The abstract is less informative. Please elaborate and revise it more precisely.
Response 1: Thank you for your comment. In the new version of the MS, we revised the abstract accordingly.
- Point 2: Please add a few illustrations, such role of biofilm in chronic infections, the mechanism of diseases, and preventive strategies.
Response 2: Following the Reviewer suggestions, we inserted a new Figure 3 that shows the polymicrobial biofilm in LCWB model and new innovative strategies.
- Point 3: Line 28, section 1 should be the introduction. What are nos 1-4? Please revise and update this section with additional pieces of information in brief also, such as i) biofilm, mechanism, its regulations (quorum sensing) in microbial pathogens, and antibiotics resistance issue for treatment i.e., FEMS Microbiology Reviews 41 (2017) 276-301; Molecules 27 (2022) 7584; ii) strategies for regulation of pathogens by inhibition such as the use of quorum sensing inhibitors and bioactive molecules such as antimicrobial agents i.e., polyhydroxyalkanoate derived materials and related compounds (Scientific Reports (2019) 9, 18160; Biotechnology Advances 37 (2019) 62-90; Bioresource Technology 326 (2021) 124737; and iii) few examples of chronic wound pathogens such as P. aeruginosa,S. aureus, E. faecalis and C. albicans their treatment strategies and worldwide economy loss due to such infections (Frontiers in Microbiology 13 (2022) 832919; Angewandte Chemie International Edition 61 (2022) e202112218).
Response 3: Following the Reviewer suggestions, we changed the title of section 1 as “Introduction”; Nos 1-4 indicate the different phases of wound healing; we change numbers 1-4 with letters A-D.
Following the Reviewer suggestions, we inserted new sentences and new references in the manuscript:
[37] Hall, C. W., & Mah, T. F. (2017). Molecular mechanisms of biofilm-based antibiotic resistance and tolerance in pathogenic bacteria. FEMS microbiology reviews, 41(3), 276–301. (pag. 5, lines 217-221);
[38] Kumar, L., Patel, S. K. S., Kharga, K., Kumar, R., Kumar, P., Pandohee, J., Kulshresha, S., Harjai, K., & Chhibber, S. (2022). Molecular Mechanisms and Applications of N-Acyl Homoserine Lactone-Mediated Quorum Sensing in Bacteria. Molecules (Basel, Switzerland), 27(21), 7584. https://doi.org/10.3390/molecules27217584 (pag. 5,6, lines 221-233);
[44] Kalia, V. C., Patel, S. K. S., Kang, Y. C., & Lee, J. K. (2019). Quorum sensing inhibitors as antipathogens: biotechnological applications. Biotechnology advances, 37(1), 68–90. https://doi.org/10.1016/j.biotechadv.2018.11.006 (pag. 7, lines 277-283);
[45] Lu, J., Cokcetin, N. N., Burke, C. M., Turnbull, L., Liu, M., Carter, D. A., Whitchurch, C. B., & Harry, E. J. (2019). Honey can inhibit and eliminate biofilms produced by Pseudomonas aeruginosa. Scientific reports, 9(1), 18160. https://doi.org/10.1038/s41598-019-54576-2 (pag. 7, lines 287,288);
[46] Di Lodovico, S., Menghini, L., Ferrante, C., Recchia, E., Castro-Amorim, J., Gameiro, P., Cellini, L., & Bessa, L. J. (2020). Hop Extract: An Efficacious Antimicrobial and Anti-biofilm Agent Against Multidrug-Resistant Staphylococci Strains and Cutibacterium acnes. Frontiers in microbiology, 11, 1852. https://doi.org/10.3389/fmicb.2020.01852 (pag. 7, lines 287,288);
[74] Darvishi, S., Tavakoli, S., Kharaziha, M., Girault, H. H., Kaminski, C. F., & Mela, I. (2022). Advances in the Sensing and Treatment of Wound Biofilms. Angewandte Chemie (International ed. in English), 61(13), e202112218. https://doi.org/10.1002/anie.202112218 (pag. 15, lines 506-511);
[88] Kalia, V. C., Singh Patel, S. K., Shanmugam, R., & Lee, J. K. (2021). Polyhydroxyalkanoates: Trends and advances toward biotechnological applications. Bioresource technology, 326, 124737. https://doi.org/10.1016/j.biortech.2021.124737 (pag. 17, lines 581-588).
- Point 4: The explanation in the citations should be more quantitatively elaborated in the text i.e, Table 1-based citations.
Response 4: According to the Reviewer comment, we insert new details in the paragraph 4.2 (see pag. 10-13, lines 413-415, 432-434, 441-443, 454-457) and in Table 1.
- Point 5: Table 1 can be extended with recent citations and brief quantitative data on inhibition.
Response 5: According to the Reviewer suggestion, we inserted new details in the Table 1 and a recent data.
- Point 6: Please provide additional illustrations on the promising therapies in chronic wounds and their significant key points and disadvantages.
- Point 7: Please add a new section about major challenges in chronic wound pathogens, especially antibiotic resistance, and potential strategies to counter them i.e., uses of quorum sensing inhibitors.
Response: According to the referee suggestion, we inserted a new figure in the new version of the manuscript (figure 3).
Reviewer 4 Report
This review manuscript describes LCWB model for its usefulness to research and clinical practice. Although authors' effort may be acknowledged and the manuscript itself is well written, this manuscript has not yet been well organized as a review paper, and there may be a concern, as explained below.
1. Structure of contents in this manuscript is not well prepared This manuscript seems to be a part of a textbook. For example, the first part starts with "1. Normal Wound Healing Process and Chronic Wound". This is not suitable for a review article. Authors should first show introduction of the wound healing and idea how and why they wanted to write the LCWB model, what is the significance and what will be evident after readers read this article. Intention and objective of this review should be clearly shown in the first part.
2. It is questionable whether this review is relevant to "molecular science". Any medical science is related to molecular phenomenon, but the main topic in this review is not molecular matter. This reviewer judge that this manuscript does not match the aim and scope of this journal (molecular biology, molecular medicine).
Author Response
RESPONSE TO REVIEWER 4 COMMENTS
This review manuscript describes LCWB model for its usefulness to research and clinical practice. Although authors' effort may be acknowledged and the manuscript itself is well written, this manuscript has not yet been well organized as a review paper, and there may be a concern, as explained below.
Point 1: Structure of contents in this manuscript is not well prepared This manuscript seems to be a part of a textbook. For example, the first part starts with "1. Normal Wound Healing Process and Chronic Wound". This is not suitable for a review article. Authors should first show introduction of the wound healing and idea how and why they wanted to write the LCWB model, what is the significance and what will be evident after readers read this article. Intention and objective of this review should be clearly shown in the first part.
Response 1: Thank you for the suggestion. According to the Reviewer comments, we modified the title and rearranged extensively the MS to make the text consistent to the title and to be more pertinent to the proposed topic of the issue (pag 1, lines 2-5)
Point 2: It is questionable whether this review is relevant to "molecular science". Any medical science is related to molecular phenomenon, but the main topic in this review is not molecular matter. This reviewer judge that this manuscript does not match the aim and scope of this journal (molecular biology, molecular medicine).
Response 2: Thank you for the comment. In the new version of the MS, we better explained the aim of the Review and the importance of the biofilms in chronic wounds focalizing the attention on the role of the LCWB that is a 3D model. The aim of the Review is in line with the special issue of S: Molecular Microbiology, SI: Microbial Biofilms and Antibiofilm Agents 3.0 that we chose for the submission. In the new version of the MS, we inserted sentences regarding the 3D gradient of the LCWB. (pag.3, lines 118-123, pag.8, lines 323-328).
Round 2
Reviewer 1 Report
The revised version of the manuscript is much better, including the change of title.
However, authors should correct minor errors throughout the manuscript (e.g. italics, punctuation, typos, table organization, as well as names of microorganisms - full name only once, as entered).
I also recommend breaking the main body (especially Chapter 6) into smaller sub-chapters.
Author Response
RESPONSE TO REVIEWER 1 COMMENTS
Point 1: The revised version of the manuscript is much better, including the change of title.
However, authors should correct minor errors throughout the manuscript (e.g. italics, punctuation, typos, table organization, as well as names of microorganisms - full name only once, as entered).
Response 1: According to the Reviewer comments, we corrected the errors in the entire MS and in the Table.
Point 2: I also recommend breaking the main body (especially Chapter 6) into smaller sub-chapters.
Response 2: In the new version of the MS, we divided the chapter 6 in small sub-sections.
Reviewer 3 Report
Accept as is
Author Response
thank you for your comments
Reviewer 4 Report
Revision has been well modified.
Author Response
thank you for your comments